# The Diversity of Developmental Age Gynecology—Selected Issues

**DOI:** 10.3390/pediatric17050091

**Published:** 2025-09-08

**Authors:** Ewa Majcherek, Justyna Jaskulska, Michalina Drejza, Katarzyna Plagens-Rotman, Karina Kapczuk, Witold Kędzia, Maciej Wilczak, Magdalena Pisarska-Krawczyk, Małgorzata Mizgier, Justyna Opydo-Szymaczek, Julia Linke, Małgorzata Wójcik, Grażyna Jarząbek-Bielecka

**Affiliations:** 1University Clinical Hospital in Poznań, 60-245 Poznań, Poland; 2Higher School of Strategic Planning in Dąbrowa Górnicza, 41-300 Dabrowa Gornicza, Poland; 3Department of Obstetrics and Gynaecology, Cambridge University Hospitals NHS Foundation Trust, Cambridge CB2 0QQ, UK; 4Division of Gynaecology, Department of Gynaecology, Poznan University of Medical Sciences, 61-758 Poznan, Poland; plagens.rotman@gmail.com (K.P.-R.);; 5Klinika Zdrowia Matki i Dziecka i Minimalnie Inwazyjnej Ginekologii Operacyjnej, Department of Maternal and Child Health and Minimally Invasive Operative Gynecology, Karol Marcinkowski University of Medical Sciences, 60-535 Poznań, Poland; 6Department of Nursing, The President Stanislaw Wojciechowski Calisia University, 62-800 Kalisz, Poland; 7Department of Sports Dietetics, Chair of Dietetics, Faculty of Health Sciences, Poznan University of Physical Education, Królowej Jadwigi 27/39, 61-871 Poznań, Poland; mizgier@awf.poznan.pl; 8Department of Pediatric Dentistry, Poznan University of Medical Sciences, 70 Bukowska Street, 60-812 Poznan, Poland; 9Florian Ceynowa Specialist Hospital, 84-200 Wejherowo, Poland; 10Department of Physiotherapy, Faculty of Physical Culture in Gorzów Wielkopolski, Poznań University of Physical Education, 61-871 Gorzów Wielkopolski, Poland; m.wojcik@awf-gorzow.edu.pl

**Keywords:** sex education, pediatric and adolescent gynaecology, adolescents’ health

## Abstract

**Background/Objectives:** Pediatric and adolescent gynaecology addresses the distinct developmental needs of the reproductive systems of young patients. Diagnosing and treating gynaecological issues in this age group are challenging due to overlapping symptoms and the developmental stage. This study aimed to identify common gynecological issues based on retrospective analysis of medical documentation from the Developmental Gynecology and Sexology Laboratory of the Gynecology Clinic, Department of Gynecology, Poznan University of Medical Sciences (UMP) from the years 2012–2023. **Methods:** The study involved 4942 patients under 18 years old. Medical records from the years 2012–2023 were analyzed, focusing on the most frequent diagnoses. Statistical analyses were performed using StatSoft STATISTICA PL 10 software, with a significance threshold of *p* < 0.05. **Results:** The most frequent diagnosis was pelvic pain syndrome (77.8%), followed by androgenization syndromes (13.2%). While the number of admissions remained stable over the years (r = 0.131, *p* > 0.05), there was a significant increase in the percentage of androgenization syndromes (*p* = 0.0040) and a decrease in pelvic pain syndrome cases (*p* = 0.0018). Other conditions such as eating disorders and psychosexual issues were also prevalent, highlighting the need for a multidisciplinary approach. **Conclusions:** The analysis indicates a shift in adolescent gynaecological diagnoses over time, with pelvic pain syndrome decreasing and androgenization syndromes increasing. The findings underline the importance of specialised, multidisciplinary care and further research to adapt diagnostic and therapeutic strategies to the changing landscape of pediatric gynaecology.

## 1. Introduction

Developmental medicine is a specialised field of clinical and theoretical medicine that investigates both the regularities and disruptions in somatic and psychological development from birth to maturity (ages 0 to 18) [1]. Within this context, pediatric and adolescent gynaecology emerges as a distinct discipline, reflecting the unique anatomical and physiological characteristics of the reproductive system before adulthood, as well as the myriad disorders typical of this developmental stage [2]. Operators face significant challenges in implementing recommended procedures and selecting appropriate active substances and dosages [3]. Various medical conditions can present with overlapping symptoms, complicating diagnosis and treatment. For instance, inflammatory conditions of the vulva, skin disorders, injuries, and the potential effects of sexual harassment may present with overlapping symptoms. It is important to remember that gonorrhea, chlamydia, and HPV also affect young individuals and can present with a wide range of symptoms, from a latent form or asymptomatic course to dysuric symptoms or classic signs of reproductive tract infections [4].

Monitoring the health of patients under 18 is paramount, starting from the neonatal period and continuing through full maturity. In addition to the intrinsic value of gynaecological examinations, consultations during childhood aim to cultivate a culture of regular preventive health practices. The American College of Obstetricians and Gynecologists recommends that the initial reproductive health visit occur between ages 13 and 15, contingent upon the absence of prior health issues [5].

The diagnostic and therapeutic approaches for younger patients significantly differ from those for adult women. Gynaecologists must be acutely aware of the distinct health concerns that may arise for girls and adolescents at various developmental milestones. In pediatric and adolescent gynaecology, eating disorders such as anorexia and bulimia, approximately 95% of first-time eating disorder cases are diagnosed before age 25, often lead to hormonal imbalances and issues with the development of the reproductive system [6]. Conditions such as polycystic ovary syndrome (PCOS), affecting 6% to 15% of girls and young women, can cause excess androgen production and disrupt normal puberty development. PCOS is also challenging to assess, as it is both underdiagnosed and overdiagnosed, which may be due to the difficulty of making an accurate diagnosis in young individuals [7,8]. Psychosexual development problems, including those arising from sexual abuse, are estimated to affect 10% to 20% of young girls, often resulting in long-term emotional and psychological consequences [9]. Breast health issues, osteoporosis prevention, and counselling on pregnancy planning and contraception are critical components of health education [10,11]. Lastly, pelvic pain syndrome affects many adolescent girls and is a common cause of chronic pain, significantly impacting their quality of life [12].

The choice of therapeutic methods is also an important aspect. Modern surgery continues to make significant progress, particularly in the field of minimally invasive procedures in comparison with traditional laparotomy. Laparoscopy is now widely used for a variety of conditions, ranging from endometriosis and uterine horn remnants to uterine mass excisions. The benefits of these techniques are numerous, including reduced postoperative pain, improved wound healing, greater patient comfort, and better cosmetic outcomes. To ensure optimal care for patients undergoing such procedures, early diagnosis of their health issues and prompt recognition of symptoms by medical professionals are crucial [3].

There is a compelling need to increase the availability of child-friendly clinics and train specialists with expertise in this field. Additionally, enhancing educational initiatives aimed at improving knowledge regarding pediatric reproductive health conditions is essential.

## 2. Materials and Methods

The retrospective study included 4942 underage patients of developmental age whose medical records from the Developmental Age Gynaecology Clinic at the Gynaecology and Obstetrics Hospital in Poznań were analysed. Inclusion criteria were obtained by being female patients under 18 years of age treated at the clinic between 2012 and 2023. Exclusion criteria included patients whose medical documentation was incomplete or if there was duplication of patient records.

The analysis focused on the reasons for patient visits, diagnoses made during treatment, and final diagnoses. Multiple diagnoses did not result in the exclusion of the patient from the study; however, only the final diagnoses, even if multiple, were considered for analysis. Diagnoses were established based on patient history, laboratory analyses, imaging studies, and physical examination. Methods were selected individually by specialists, guided by their clinical judgment, experience, and established practice preferences. Ethical standards were strictly upheld throughout the study. No sensitive or personally identifiable data were included in the analysis, and the extraction of information from medical archives was conducted by the hospital’s internal anonymisation protocols to ensure full patient confidentiality.

Statistical analyses were performed using the StatSoft STATISTICA PL 10 software package. A *p*-value of <0.05 was considered statistically significant.

## 3. Results

1.The number of admissions from 2012 to 2022 showed no significant upward or downward trend (r = 0.131, *p* > 0.05). The lowest number of admissions was recorded in 2020 (*n* = 299), possibly due to the impact of the COVID-19 pandemic, while the highest number occurred in 2022 (*n* = 612), indicating a potential backlog of cases being addressed post-pandemic (Table 1, Figure 1). No trend in the number of admissions was observed in the years 2013–2023 (no significant correlation, *p* > 0.05). Trend analysis was performed using Pearson’s linear correlation coefficient significance test (r values) and Spearman’s rank correlation coefficient significance test (Rs values).2.Between 2012 and 2023, in terms of diagnoses, patients were most frequently admitted due to pelvic pain syndrome (77.8% of all diagnoses; Figure 2). The second most common diagnosis was androgenization syndrome (13.2% of all diagnoses; Figure 3).

During the analysis, the use of the N94 diagnosis proved to be problematic, as it currently includes various conditions such as:N94.0—Mittelschmerz (Intermenstrual pain)N94.1—Dyspareunia (Painful sexual intercourse)N94.2—VaginismusN94.3—Premenstrual tension syndromeN94.4—Primary dysmenorrheaN94.5—Secondary dysmenorrheaN94.6—Unspecified dysmenorrheaN94.8—Other specified conditions associated with female genital organs and the menstrual cycleN94.9—Unspecified condition associated with female genital organs and the menstrual cycle

In our study, the focus was on identifying pain syndromes in general. Therefore, detailed diagnoses such as N94.1 (Dyspareunia) and N94.2 (Vaginismus) were not connected to this group.

**Table 1 pediatrrep-17-00091-t001:** Number and Frequency of Admissions by Diagnosis in the Years 2012–2023.

Pelvic Pain Syndrome	2012	2013	2014	2015	2016	2017	2018	2019	2020	2021	2022	2023	Total
Androgenization Syndromes	396	406	372	370	342	266	219	289	188	262	430	307	3847
Eating Disorders	9	6	2	4	6	45	83	117	84	118	118	60	652
Psychosexual Development Issues	19	9	28	6	1	15	25	35	22	30	28	20	238
Other	8	35	22	26	4	5	4	3	5	2	36	42	192
Total	4	3	1	1	1	0	0	0	0	2	0	1	13
	436	459	425	407	354	331	331	444	299	414	612	430	4942

**Figure 1 pediatrrep-17-00091-f001:**
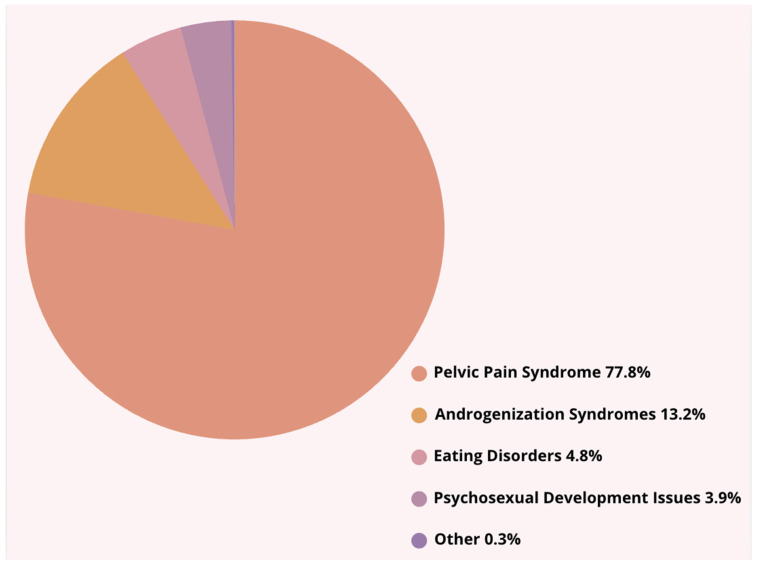
Percentage distribution of diagnoses from 2012 to 2023.

**Figure 2 pediatrrep-17-00091-f002:**
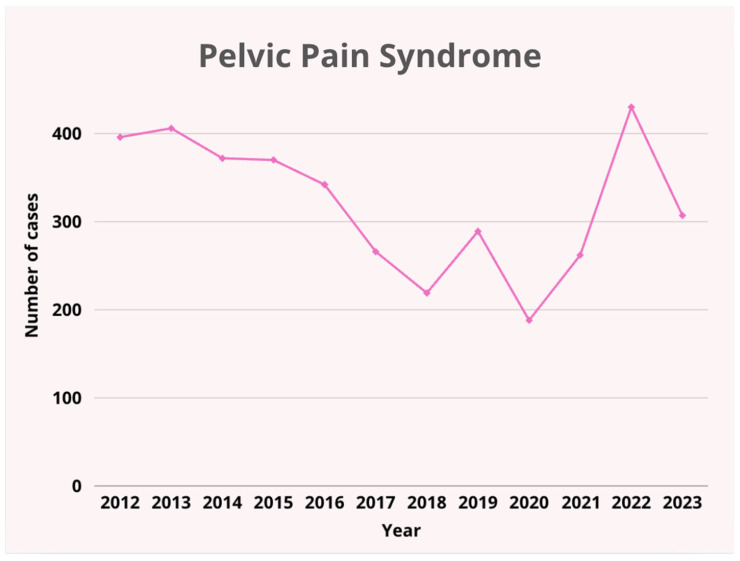
Trend in the prevalence of pelvic pain syndrome over the years.

**Figure 3 pediatrrep-17-00091-f003:**
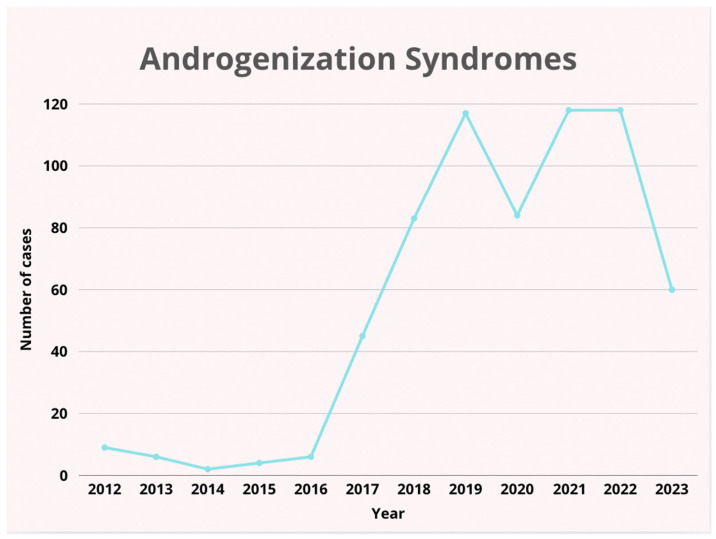
Trend in the prevalence of androgenization syndromes over the years.

3.Significant Correlations Between Years and the Percentage Share of Diagnoses
(1)Androgenization Syndromes (*p* = 0.0040)—a strong positive correlation (Rs = 0.762). In the following years, the percentage share of this diagnosis among all diagnoses increased (upward trend).(2)Preventive Care, Issues Related to the Physiology and Pathology of the Mammary Glands, Osteoporosis, Pregnancy Planning Counselling, Contraception (*p* = 0.0471)—a moderate negative correlation (Rs = −0.582). Over the years, there was a decrease in the percentage share of this diagnosis among all diagnoses (downward trend).(3)Pelvic Pain Syndrome (*p* = 0.0018)—a strong negative correlation (Rs = −0.798). In the following years, the percentage share of this diagnosis among all diagnoses decreased (downward trend).4.Diverse Conditions: While pelvic pain syndrome and androgenization syndromes dominate the diagnostic landscape, other conditions such as eating disorders, psychosexual issues, and breast pathology also play an important role. This underscores the need for a multidisciplinary approach in diagnosis and treatment, particularly in managing hormonal, psychosexual, and metabolic disorders.

## 4. Discussion

The data collected illustrate the evolving health issues faced by adolescents over the years. A similar study conducted at the same clinic between 1994 and 2001 highlights significant differences in diagnoses and treatment practices. Notably, the incidence of androgenization syndromes (5.3%) and pelvic pain syndrome (3.9%) was lower than, of vulvovaginitis and menstrual disorders, which were the leading concerns at that time [13]. The classification of “pelvic pain syndrome,” which demonstrated a prevalence of nearly 78% in our study, approximately 20 years later, may also raise questions regarding diagnostic specificity and the potential for more accurate diagnostic approaches. Additionally, the teenage pregnancy rate in Poland has significantly declined, from 35 per 1000 girls in 1994 to 9.6 per 1000 in 2020, reflecting changing focuses in youth counselling services [14,15].

The observed increase in the androgenization syndromes in our study aligns with global trends in the diagnosis of this condition. This rise may be attributed to a range of specific underlying disorders, most notably polycystic ovary syndrome (PCOS), which is considered the most common gynaecological disease or obesity. Additionally, greater awareness among patients and healthcare providers, as well as advancements in diagnostic methods, likely contribute to this phenomenon [7,16,17,18,19].

A study conducted in the UK between 1997 and 2014 also identifies common reasons for visits to pediatric and adolescent gynaecology services, such as menstrual disorders (30.6%), abdominal pain (24.5%), and reproductive system issues (20.4%) [20]. A different UK study reported statistics, with common presentations including vulvovaginitis (18%), labial adhesions (14%), abdominal pain (7%), and precocious puberty (6%) [21]. Due to the differences in symptoms and the complexity of the ICD-10 classification, as mentioned in the results, it is challenging to appropriately categorise selected medical cases. A deeper exploration of this topic is essential, considering the complexity of the problem in children and adolescents. Our study does not discuss several important limitations that may affect the interpretation of the findings. These include potential coding errors and diagnostic ambiguities, particularly, as previously mentioned, those arising from the application of ICD-10 classification, which may vary between clinicians or institutions. Additionally, institutional practices influencing patient selection and case mix could have introduced systematic bias into the dataset. Study limitations also include the absence of detailed demographic and clinical data on the patients, which prevents a comprehensive characterization of the study population and identification of factors influencing gynecological conditions in girls. The lack of information on patient age at first visit, socioeconomic status, comorbidities, and specifics of physical and psychological development limits the depth of trend analysis. These data gaps affect result interpretation and restrict the generalizability of findings to the broader population of girls of developmental age, potentially hindering the application of results in healthcare planning. Analysis is limited to correlations observed over time, without adjusting for confounding variables or investigating underlying causal relationships.

In another study by the American College of Obstetricians and Gynaecologists (ACOG) Collaborative Ambulatory Practice Network, obstetrician-gynaecologists who typically care for adolescent patients primarily focused on reproductive health services for adolescent patients, such as contraception, menstrual management, and HPV immunisations. While these specialists frequently provided reproductive health services to adolescents, a notable portion (55.2%) also offered primary care services, including immunisations. These findings highlight regional variations in health concerns among young patients [22].

In some countries, primary care providers, including family physicians, play a significant role in guiding young individuals through essential health matters, including those related to Differences of Sexual Development (DSD) and menstrual health management. Their responsibilities often encompass contraceptive counselling, screening for sexually transmitted infections, and addressing psychosocial aspects, such as body image and autonomy in health-related decision-making. Numerous global organisations, such as the Royal College of Obstetricians and Gynaecologists (RCOG) and the American Academy of Paediatrics, have developed guidelines and courses to equip these primary care physicians better to manage basic reproductive health issues within the general practice setting [23,24,25,26]. Expanding the competencies of primary care providers in this field could not only relieve secondary care specialists, who traditionally handle such concerns in countries like Poland, but also enhance awareness and early detection of reproductive health conditions. The primary care setting is often the most comfortable environment for young patients, fostering trust and strengthening the physician-patient relationship from an early stage in their development [27].

On the other hand, obstetrician-gynaecologists in some societies function as a type of primary care professional, which stems from their comprehensive role in women’s health care. Many patients view their obstetrician-gynaecologist as their primary care physician, particularly for preventive health and routine care needs. From another medical perspective, the data reveals important insights: among 935 respondents (with a 55% response rate), physicians estimated that 37% of their private, nonpregnant patients depend on them for routine primary care. Additionally, around 22% of these physicians indicated that they required further training in primary care, particularly in areas such as metabolism and nutrition, as well as in dermatological, cardiovascular, and psychosexual disorders [28]. Because obstetrician-gynaecologists frequently perform primary care duties, they often serve as the first point of contact within the healthcare system. However, as healthcare reforms pose potential barriers to patient access to obstetrician-gynaecologist services, there is a pressing need for changes in training and practice models. Enhancing obstetrician-gynaecologist education to further develop skills in general preventive care and the management of non-gynecologic conditions will be crucial in maintaining their role as trusted primary care providers for women [29,30]. This is particularly important considering that even in European countries, an integrated approach to women’s health is not always available or accessible to patients [31].

Holistic approaches that include educational initiatives related to normal and abnormal development are critical. Properly attending to a child’s development and education also has a significant impact on their safety, particularly in the context of preventing sexual abuse [32,33]. Promoting health education and preventive strategies in schools is vital for increasing awareness of bodily autonomy and health, enabling adolescents to recognise alarming symptoms. Consequently, providing reliable, comprehensive, accessible, and evidence-based sexual and reproductive health education is essential for this age group.

Currently, a significant majority of patients are diagnosed with pelvic pain syndrome, which can often be confused with other conditions [34,35]. It is important to emphasise that chronic pelvic pain in adolescents may have a broad range of non-gynaecological causes, many of which are closely linked to lifestyle and psychosocial conditions. Factors such as bullying, chronic stress, and psychiatric disorders—including depression, anxiety, and even attention-deficit/hyperactivity disorder (ADHD)—can manifest with somatic symptoms that mimic gynaecological pain syndromes [36,37,38,39]. These psychosocial stressors may not only present as pelvic discomfort but can also exacerbate existing gynaecological conditions such as dysmenorrhea. Therefore, a comprehensive, multidisciplinary approach to diagnosis and management—one that integrates psychological and social assessment alongside gynaecological evaluation—is essential in effectively addressing pelvic pain in young individuals. It is imperative to enhance awareness among paediatricians, family physicians, and the broader medical community regarding the importance of comprehensive medical histories and thorough examinations for gynaecological conditions. Additionally, directing adolescents to specialised pediatric and adolescent gynaecology clinics with skilled professionals is crucial for addressing their concerns [40,41].

Gynaecological care is a fundamental aspect of health protection for women and girls. From adolescence, marked by numerous hormonal changes, to late adulthood, care should be tailored to individual age and health status. Collaboration with professionals from other specialities, including dentists, dietitians, and psychologists, is essential [13,42]. The data analysis from the clinic underscores the necessity for a multifaceted approach in modern developmental gynaecology. The complexity of gynaecological issues highlights the need for advanced diagnostic and therapeutic methods, considering women’s sexual health and the psychosexual development of girls. There is a significant demand for healthcare professionals specialising in developmental medicine, particularly in gynaecology, sexology, and reproductive health [43,44].

Minors constitute a distinct category of patients, necessitating the creation of an open and comfortable atmosphere within clinical settings. Effective communication using straightforward language and clear explanations of procedures is crucial. Employing inclusive language and actively involving adolescent patients in their therapeutic processes is essential, particularly for those aged 16 and older, who must legally consent to medical interventions [45,46].

The presence of accompanying individuals, such as a parent or guardian, can foster a sense of security during medical visits [47]. For older adolescents, considering a caregiver’s presence behind a privacy screen can ensure intimacy while alleviating embarrassment [48].

Adolescence is a period filled with myths and uncertainties that young girls may struggle to discuss with family members. Therefore, consulting a specialist in pediatric and adolescent gynaecology provides valuable knowledge, aids in understanding their bodies, and fosters the habit of healthcare, transforming gynaecological examinations from a source of fear or shame into a routine aspect of healthcare.

## 5. Conclusions

The analysis indicates a dynamic shift in adolescent gynaecological diagnoses over time, with changes in the prevalence of different diagnostic groups. While the incidence of pelvic pain syndrome has decreased, it remains the most significant condition that requires further investigation in this population. At the same time, disorders such as androgenization syndromes are becoming increasingly common, highlighting the need for greater attention to previously less frequent diagnoses like the one mentioned above. Proper adolescent care improves adult health outcomes and enhances the success of future gynaecological treatment.

These findings emphasise the growing necessity for young individuals to seek specialised care in gynaecological clinics and the importance of a multidisciplinary approach to diagnosis and treatment. Regular studies should be conducted at consistent time intervals to monitor trends, identify emerging conditions, and guide primary healthcare providers in recognising and addressing the most pressing issues in pediatric and adolescent gynaecology.

## Data Availability

Data is contained within the article. The original contributions presented in the study are included in the article; further inquiries can be directed to the corresponding author.

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
