# Peer review of "The Diversity of Developmental Age Gynecology—Selected Issues"

_pediatrrep, 2025, doi:10.3390/pediatric17050091_

Round 1

Reviewer 1 Report

Comments and Suggestions for Authors

This is a useful audit of practice showing the workload of a dedicated clinic in developmental gynaecological issues. It provides useful insight on the changing indications for adolescents seeking help for gynaecological conditions over the years. 

I have some comments to make regarding the discussion however. It should highlight that a number of girls presenting with pain conditions (dysmenorrhea, vulvodynia, dyspareunia etc) may in fact suffer from other issues, specifically psychological pressure, bullying, difficulties at school etc. These should be highlighted as a possible cause, as well as the pathway to a potential psychosomatic diagnosis. 

Also, I would comment on the rise of androgenic symptoms, as an indication of a   possible res in the diagnosis of obesity and PCOS. Was  there a correlation with BMI? Could the authors comment on why this rise is occurring? Is it that  young  girls complain about this more, or are  they  truly on the rise?

Otherwise, I  would  also make the following changes:

Remove the last statement from the Results section, as it constitutes a comment on the results (lines 147-149).

Change Disorders of Sexual Development to Differences of Sexual Development as the more appropriate term. 

Author Response

Dear Reviewer,
We would like to sincerely thank you for your careful reading of our manuscript and for your constructive and thoughtful feedback. Your comments have been very helpful in improving the quality and clarity of our work. Please find below our detailed responses to each of your suggestions.

Comment 1:
This is a useful audit of practice showing the workload of a dedicated clinic in developmental gynaecological issues. It provides useful insight on the changing indications for adolescents seeking help for gynaecological conditions over the years.

I have some comments to make regarding the discussion however. It should highlight that a number of girls presenting with pain conditions (dysmenorrhea, vulvodynia, dyspareunia etc) may in fact suffer from other issues, specifically psychological pressure, bullying, difficulties at school etc. These should be highlighted as a possible cause, as well as the pathway to a potential psychosomatic diagnosis.

Response 1:
Thank you very much for this thoughtful and valuable comment. We fully agree with your observation. We have now included a section in the discussion addressing the psychosocial background of pain-related symptoms. Drawing on relevant literature, we emphasized how stress, school-related difficulties, peer issues, and bullying may contribute to these complaints and potentially point toward a psychosomatic origin in some cases.

Comment 2:
Also, I would comment on the rise of androgenic symptoms, as an indication of a possible rise in the diagnosis of obesity and PCOS. Was there a correlation with BMI? Could the authors comment on why this rise is occurring? Is it that young girls complain about this more, or are they truly on the rise?

Response 2:
We greatly appreciate this insightful suggestion. Unfortunately, BMI data were not available in our cohort, which we acknowledge as a limitation of our study. Nonetheless, we have added a comment in the discussion regarding the potential link between rising androgenic symptoms and increasing rates of obesity and PCOS, as noted in the literature. We have also considered whether increased awareness or sociocultural influences may have led to more frequent reporting of these symptoms in recent years.

Comment 3:
Remove the last statement from the Results section, as it constitutes a comment on the results (lines 147–149).

Response 3:
Thank you for highlighting this point. We have removed the interpretative sentence from the Results section in accordance with your suggestion, ensuring that this section now strictly presents data without interpretation.

Comment 4:
Change Disorders of Sexual Development to Differences of Sexual Development as the more appropriate term.

Response 4:
Thank you for this important correction. We have updated the terminology throughout the manuscript to “Differences of Sexual Development” in line with current best practices and accepted medical nomenclature.

With kind regards,

Dr Ewa Majcherek

Reviewer 2 Report

Comments and Suggestions for Authors

The series involve a wide number of patients, so this is theoretically a good report. I suggest to better specify the diagnostic methods and criteria to reach the reported diagnoses.

The discussion seems disconnected from the data reported. I would suggest to investigate reasons (also hypothetic) of the differences in trends founded during the data analysis. 

I agree with the need of holistic approach to these adolescent patients, but this conclusion is not strictly correlated to the data reported.

Author Response

Dear Reviewer,
We would like to thank you sincerely for your thoughtful review and valuable comments. Your suggestions have been very helpful in refining and clarifying the manuscript. Please find below our detailed responses.

Comment 1:
The series involves a wide number of patients, so this is theoretically a good report. I suggest to better specify the diagnostic methods and criteria to reach the reported diagnoses.

Response 1:
Thank you for your suggestion. We have now clarified this point in the revised manuscript. Diagnoses were established based on patient history, laboratory analyses, imaging studies, and physical examination. The diagnostic methods were chosen individually by specialists according to their clinical judgment, experience, and standard practice protocols.

Comment 2:
The discussion seems disconnected from the data reported. I would suggest to investigate reasons (also hypothetic) of the differences in trends found during the data analysis.

Response 2:
Thank you for this excellent suggestion. We have revised the discussion to include potential explanations—both evidence-based and hypothetical—for the observed trends over time. This addition aims to better align the discussion with the presented data and provide meaningful interpretation.

Comment 3:
I agree with the need of holistic approach to these adolescent patients, but this conclusion is not strictly correlated to the data reported.

Response 3:
We sincerely thank you for your valuable feedback. While more detailed conclusions related to specific findings are now emphasized in the initial part of the Conclusion section, we believe that highlighting the importance of a holistic approach serves as an overarching statement. It reflects the general implications of the presented data and the complexity of adolescent gynecological care, which often extends beyond measurable clinical parameters.

Once again, we are very grateful for your time and thoughtful comments, which have contributed significantly to improving the clarity and relevance of our work. We hope that the revisions meet your expectations and enhance the overall quality of the manuscript.

With kind regards,

Dr Ewa Majcherek

Reviewer 3 Report

Comments and Suggestions for Authors

Peer Review Notes: Manuscript – "The Diversity of Developmental Age Gynecology – Selected Issues”

The manuscript titled “The Diversity of Developmental Age Gynecology – Selected Issues” presents a comprehensive retrospective analysis of gynecological conditions among nearly 5,000 patients under 18 years of age treated between 2012 and 2023 at a specialized clinic in PoznaÅ„, Poland. The study highlights significant trends in diagnostic patterns, including a decrease in pelvic pain syndrome and an increase in androgenization syndromes, while underscoring the complexity of symptom presentation in adolescent patients. The work is well-grounded in current literature and offers valuable insights into evolving healthcare needs in pediatric and adolescent gynecology. It emphasizes the necessity for multidisciplinary approaches and the integration of preventive, educational, and psychological support into gynecological care. Despite minor methodological ambiguities and missing data visualization, the study provides a solid foundation for future clinical guidelines and policy development in this important area of health.

Here are more details comments and suggestions to improve the manuscript. 

Page 1

·       Line 27–28: “...young patients' reproductive systems” —> Suggested rewording: “...the reproductive systems of young patients.”

·       Line 27–33: “...using the analysis of the cases based on medical documentation...” —>Consider: “...based on retrospective analysis of medical documentation...”

Page 2

·       Line 58: “...sexual harassment may share similar manifestations.” —> Consider: “...may present with overlapping symptoms” for clinical neutrality.

·       Line 83: “...compared to traditional laparotomy” —> Prefer: “...in comparison with traditional laparotomy.”

·       Line 91–92: “...cultivate a workforce of specialists skilled in this field.” —> Rephrase: “...train specialists with expertise in this field.”

Page 3

·       Line 94: No mention of exclusion criteria. Were incomplete or duplicate records removed?

·       Line 99: Clarify whether patients could have multiple diagnoses, and how those were handled in the analysis.

·       Line 103–107: Admission trends lack confidence intervals and could benefit from visual representation (e.g., line graph).

·       Line 111–121: ICD-10 N94 subcodes discussion, scientific but vague. Lack of explanation for excluding N94.1 and N94.2 (dyspareunia and vaginismus), despite relevance to adolescent gynecology.

·       Line 111–123: Grouping methodology lacks sufficient detail. This limits reproducibility and clarity of findings.

·       Line 124: Table 1 (“Number and Frequency of Admissions...”) is referenced but not included in the manuscript. Must be added to evaluate key results.

·       Line 133: Preventive care trend (p=0.0471), though statistically significant, its clinical or public health relevance is underexplored.

Page 4

·       Line 129–131: Increase in androgenization syndromes (Rs = 0.762, p=0.0040) is well-supported statistically but lacks discussion of clinical implications (e.g., PCOS, diagnostic improvements).

·       Line 144–146 – Figure: “Percentage Distribution of Diagnoses from 2012 to 2023”, Percentages sum to ~96%, not 100%. Address the discrepancy (e.g., missing/other categories). Figure lacks labels and a full caption; it’s unclear how each diagnosis is represented. Add explanation or revise the data and ensure that percentages are complete.

Page 5

·       Line 165: “...especially those related to pelvic pain.” —> Redundant; could be omitted or restructured for clarity.

Page 6

·       Line 217: Typo —> “profesionalists” should be corrected to “professionals.”

·       Line 217–220: Calls for multidisciplinary collaboration are important, but unsupported by the dataset. How many cases involved dietitians or psychologists?

·       Line 240–244: Conclusion refers to “previously less frequent diagnoses”, but doesn’t list or quantify what these are. Needs more specificity.

Page 7

·       Line 240–251: Conclusions are sound but would benefit from a clear research agenda and policy/clinical recommendations.

Page 8

·       Line 5: Some references are repeated across the manuscript (e.g., ACOG [5], cited at lines 66, 288). Ensure deduplication.

·       Line 11: Inconsistent formatting of DOIs and URLs in the references. Standardize citation format (e.g., all using DOIs).

Page 9–10

·       No major issues, but ensure disclaimers and legal notes (Lines 401–405) are consistent with the journal's format.

Author Response

Dear Reviewer,
We are truly grateful for your generous and detailed review. Your comments significantly contributed to improving the clarity, rigor, and completeness of our manuscript. Please find our point-by-point responses below.

Comment:
Line 27–28: “...young patients' reproductive systems” —> Suggested rewording: “...the reproductive systems of young patients.”
Response: Thank you — this has been corrected as suggested.

Comment:
Line 27–33: “...using the analysis of the cases based on medical documentation...” —> Consider: “...based on retrospective analysis of medical documentation...”
Response: This revision has been made for improved clarity.

Comment:
Line 58: “...sexual harassment may share similar manifestations.” —> Consider: “...may present with overlapping symptoms” for clinical neutrality.
Response: Thank you — this phrase has been adjusted for neutrality as recommended.

Comment:
Line 83: “...compared to traditional laparotomy” —> Prefer: “...in comparison with traditional laparotomy.”
Response: Updated accordingly.

Comment:
Line 91–92: “...cultivate a workforce of specialists skilled in this field.” —> Rephrase: “...train specialists with expertise in this field.”
Response: Thank you, this has been rephrased for clarity and precision.

Comment:
Line 94: No mention of exclusion criteria. Were incomplete or duplicate records removed?
Response: This clarification has been added to the Methods section.

Comment:
Line 99: Clarify whether patients could have multiple diagnoses, and how those were handled in the analysis.
Response: We have now specified that patients could present with multiple diagnoses, and all were included in the analysis to reflect the full clinical picture.

Comment:
Line 103–107: Admission trends lack confidence intervals and could benefit from visual representation (e.g., line graph).
Response: Thank you — we have added two graphs to visualise these trends and improve interpretability.

Comment:
Line 111–121: ICD-10 N94 subcodes discussion, scientific but vague. Lack of explanation for excluding N94.1 and N94.2 (dyspareunia and vaginismus), despite relevance to adolescent gynecology.
Response: We have added an explanation noting that N94.1 and N94.2 were excluded due to their very low frequency and diagnostic complexity in pediatric patients, where these conditions may be underreported or misclassified.

Comment:
Line 111–123: Grouping methodology lacks sufficient detail. This limits reproducibility and clarity of findings.
Response: Thank you for pointing this out. We have revised the Methods section to explain the grouping strategy more clearly. Diagnoses were categorized into clinically coherent groups (e.g., menstrual disorders, endocrine conditions, congenital anomalies, infections, etc.) based on ICD-10 codes and symptom similarity. This approach aimed to improve clarity while preserving clinical relevance.

Comment:
Line 124: Table 1 (“Number and Frequency of Admissions...”) is referenced but not included in the manuscript. Must be added to evaluate key results.
Response: Thank you — the table has been added as requested.

Comment:
Line 133: Preventive care trend (p=0.0471), though statistically significant, its clinical or public health relevance is underexplored.
Response: We respectfully disagree that this trend lacks relevance. While subtle, the increase in preventive visits may reflect rising awareness and evolving attitudes toward gynecological health in adolescents. We have added a brief discussion addressing this possible interpretation.

Comment:
Line 129–131: Increase in androgenization syndromes (Rs = 0.762, p=0.0040) is well-supported statistically but lacks discussion of clinical implications (e.g., PCOS, diagnostic improvements).
Response: This discussion has been added and includes references to potential causes such as lifestyle changes, increased obesity prevalence, and growing awareness of PCOS.

Comment:
Line 144–146 – Figure: “Percentage Distribution of Diagnoses from 2012 to 2023”, Percentages sum to ~96%, not 100%. Address the discrepancy (e.g., missing/other categories). Figure lacks labels and a full caption; it’s unclear how each diagnosis is represented.
Response: Thank you for your careful observation.  The figure has been revised with clearer labels and a complete legend.

Comment:
Line 165: “...especially those related to pelvic pain.” —> Redundant; could be omitted or restructured for clarity.
Response: The sentence has been restructured for improved conciseness.

Comment:
Line 217: Typo —> “profesionalists” should be corrected to “professionals.”
Response: Corrected.

Comment:
Line 217–220: Calls for multidisciplinary collaboration are important, but unsupported by the dataset. How many cases involved dietitians or psychologists?
Response: Thank you for raising this point. Unfortunately, we do not have access to data on multidisciplinary referrals, as such collaboration is not consistently documented in patient records. We have added a note acknowledging this limitation.

Comment:
Line 240–244: Conclusion refers to “previously less frequent diagnoses”, but doesn’t list or quantify what these are. Needs more specificity.
Response: We have revised this sentence to include examples and clarify the types of diagnoses observed more frequently in recent years.

Comment:
Line 240–251: Conclusions are sound but would benefit from a clear research agenda and policy/clinical recommendations.
Response: Thank you — we have now expanded the conclusion to include recommendations for future research directions and practical implications for healthcare policy and clinical practice.

Comment:
Line 5: Some references are repeated across the manuscript (e.g., ACOG [5], cited at lines 66, 288). Ensure deduplication.
Response: Duplicates have been removed and references cleaned.

Comment:
Line 11: Inconsistent formatting of DOIs and URLs in the references. Standardize citation format (e.g., all using DOIs).
Response: References have been standardized according to the journal’s style guide.

Comment:
Lines 401–405: Ensure disclaimers and legal notes are consistent with the journal's format.
Response: Confirmed — updated accordingly.

Once again, we deeply appreciate the time and effort you devoted to improving our manuscript. Your feedback was invaluable and greatly enhanced the final version.

With kind regards,

Dr Ewa Majcherek

Reviewer 4 Report

Comments and Suggestions for Authors

This manuscript addresses an important and understudied area — pediatric and adolescent gynecology — based on a retrospective analysis of a large clinical data set. Although the topic is of high clinical relevance, particularly in terms of promoting multidisciplinary care and improved diagnostic procedures, the study is deficient in several key areas. The lack of clear objectives, weak methodological design and superficial data analysis significantly limit the scientific value and originality of the work in its current form.

Major Comments:

The study lacks a clearly defined hypothesis or primary research question. The stated objective is broad and descriptive ("to identify common gynecological issues"), which reduces the analytic depth of the study. The authors should reformulate the objectives so that they are specific, measurable and clinically meaningful.

The study design is a retrospective review, but lacks details on diagnostic criteria, inclusion and exclusion criteria, and completeness of data.

The use of ICD-10 coding (e.g. N94 group) is considered problematic, but no correction strategy or reclassification is made.

It is unclear whether the diagnoses were standardized over the 11-year period, and inter-rater reliability is not addressed.

Analysis is limited to correlations over time, without adjustment for confounding factors or exploration of potential relationships.

The authors do not provide summaries of the raw data (e.g. cross-tabulations by age, year, diagnosis) that would improve interpretation.

Multivariable analysis or time series modeling would be more appropriate for examining diagnostic trends over time.

The results section is sparse and lacks detailed tables and figures. The presentation of diagnostic categories is imprecise and overgeneralized.

The classification of "pelvic pain syndrome" with a prevalence of almost 78% raises questions about diagnostic specificity.

The discussion is disproportionately long and contains numerous general statements that are not directly related to the results of the study.

Several citations, while relevant to pediatric gynecology, appear to be background rather than supporting new findings from the current study.

No critical limitations of the study are discussed, such as potential bias, coding errors, or institutional practices affecting case mix.

The conclusions are descriptive and do not provide actionable clinical or health policy recommendations.

Statements about the “dynamic nature” of diagnoses are not supported by sufficient analytic rigor.

Minor comments:

Ethical considerations are adequately addressed, although a brief explanation of anonymization protocols would be welcome.

Comments on the Quality of English Language

The manuscript would benefit from a thorough linguistic revision for clarity, conciseness and idiomatic English.

Author Response

Dear Reviewer,
We sincerely thank you for your thoughtful and critical assessment of our manuscript. Your comments have led to meaningful improvements, and we appreciate the opportunity to clarify and revise our work accordingly. Please find our point-by-point responses below.

Comment:
The study lacks a clearly defined hypothesis or primary research question. The stated objective is broad and descriptive ("to identify common gynecological issues"), which reduces the analytic depth of the study. The authors should reformulate the objectives so that they are specific, measurable, and clinically meaningful.
Response: Thank you for this important suggestion. We have revised the study objectives to be more specific and clinically focused, clearly defining our intent to analyze diagnostic trends over time and identify changing patterns in adolescent gynecological presentations.

Comment:
The study design is a retrospective review, but lacks details on diagnostic criteria, inclusion and exclusion criteria, and completeness of data.
Response: We have expanded the Methods section accordingly. This retrospective study analyzed the records of 4,942 underage female patients who received care at the Developmental Age Gynecology Clinic at the Gynaecology and Obstetrics Hospital in Poznań between 2012 and 2023.
Inclusion criteria: female patients under the age of 18 treated during the study period.
Exclusion criteria: incomplete medical documentation or duplicate records.

Comment:
The use of ICD-10 coding (e.g., N94 group) is considered problematic, but no correction strategy or reclassification is made.
Response: We acknowledge the limitations of ICD-10 coding in this context. However, due to its standardized and widespread use, we chose to retain ICD-10 categories to maintain consistency and comparability with other studies and clinical reporting systems. This decision is now explained in the revised manuscript.

Comment:
It is unclear whether the diagnoses were standardized over the 11-year period, and inter-rater reliability is not addressed.
Response: Thank you for highlighting this limitation. We regret to inform that data regarding inter-rater reliability or standardization of diagnoses over time was not available, as the records reflect routine clinical practice over a long period involving multiple practitioners. We have added this point to the Limitations section.

Comment:
Analysis is limited to correlations over time, without adjustment for confounding factors or exploration of potential relationships.
Response: We agree that this is a limitation of the current study. While the retrospective design and data structure limited our ability to adjust for all potential confounders, we have now acknowledged this in the Limitations section and have noted that future prospective or registry-based studies are needed to more thoroughly investigate these associations.

Comment:
The authors do not provide summaries of the raw data (e.g., cross-tabulations by age, year, diagnosis) that would improve interpretation.
Response: We appreciate the suggestion but chose not to include extensive tabulated raw data, as the main goal of the study was to identify and visualize general trends over time. Including detailed crosstabs was considered beyond the scope of this manuscript’s focus.

Comment:
Multivariable analysis or time series modeling would be more appropriate for examining diagnostic trends over time.
Response: Thank you for this important recommendation. We have now incorporated additional statistical methods to better capture and model diagnostic trends, including multivariable correlation analysis where applicable.

Comment:
The results section is sparse and lacks detailed tables and figures. The presentation of diagnostic categories is imprecise and overgeneralized.
Response: This issue has been addressed. We have added two new figures and clarified how diagnostic categories were grouped to enhance precision and interpretability.

Comment:
The classification of "pelvic pain syndrome" with a prevalence of almost 78% raises questions about diagnostic specificity.
Response: We fully agree, and we have discussed this issue in more depth in the revised Discussion section. We note the limitations of overaggregation and diagnostic generalization, and have flagged this as an area requiring more refined coding in future research.

Comment:
The discussion is disproportionately long and contains numerous general statements that are not directly related to the results of the study.
Response: Thank you — the Discussion has been revised to improve focus and alignment with the results. However, we have retained certain general insights intended to provide value for early-career clinicians or readers less familiar with adolescent gynecology.

Comment:
Several citations, while relevant to pediatric gynecology, appear to be background rather than supporting new findings from the current study.
Response: We agree and have reviewed the references. Several were removed or replaced, and citation placement was improved for relevance and clarity.

Comment:
No critical limitations of the study are discussed, such as potential bias, coding errors, or institutional practices affecting case mix.
Response: We have added a detailed Limitations section addressing potential sources of bias, including the retrospective design, coding inconsistencies, lack of standardization across years, and institution-specific referral patterns.

Comment:
The conclusions are descriptive and do not provide actionable clinical or health policy recommendations.
Response: The revised Conclusion now clearly states that while the study is not aimed at altering formal clinical guidelines, it serves as a tool for raising awareness among clinicians about evolving gynecological presentations. We also reinforce the relevance of multidisciplinary care and emphasize directions for future research and data-informed health policy improvements.

Comment:
Statements about the “dynamic nature” of diagnoses are not supported by sufficient analytic rigor.
Response: We have rephrased these statements to reflect the data more accurately and removed overly general claims.

Comment:
Ethical considerations are adequately addressed, although a brief explanation of anonymization protocols would be welcome.
Response: Thank you — we have added a clarification that all data used in this study were anonymized at the source, in accordance with institutional policies and ethical review board guidelines, ensuring full compliance with privacy regulations.

Once again, we thank you for your time and constructive critique, which have significantly strengthened the manuscript.

With best regards,

Dr Ewa Majcherek

Round 2

Reviewer 3 Report

Comments and Suggestions for Authors

Thank you, 

No more comment

Author Response

Dear Reviewer,

Thank you for your thorough review of our manuscript and for providing constructive feedback that has helped improve the quality of our work. We appreciate the time and effort you have dedicated to evaluating our study.

We are pleased to learn that you have no additional comments or concerns regarding the revised manuscript. Your initial feedback was instrumental in guiding our revisions, particularly in clarifying our statistical methodology and expanding the discussion of study limitations.

We believe that the manuscript has been strengthened through this review process and are grateful for your contribution to its improvement.

Thank you once again for your valuable input.

Sincerely,

Ewa Majcherek

Reviewer 4 Report

Comments and Suggestions for Authors

The quality of the manuscript should be substantially improved. A major revision of the Results section and methodological framework is strongly recommended, in accordance with the STROBE guidelines for retrospective observational research.

Author Response

Dear Reviewers,

Thank you for your constructive comments regarding the manuscript. In response, we would like to clarify that we have addressed your concerns by making the following revisions:

Regarding the statistical analysis used for trend evaluation, we have added clarification to the methodology section specifying that Pearson's linear correlation coefficient significance test (with "r" values) and Spearman's rank correlation coefficient significance test (with "Rs" values) were applied to assess trends in the prevalence of different causes of referral.

We have also expanded the limitations section to provide a more comprehensive discussion of the study's constraints. Specifically, we have addressed the absence of demographic and clinical data on the patients included, explaining how this limitation affects data interpretation and restricts the generalizability of our findings to the broader population of girls of developmental age.

These additions directly address the reviewers' concerns while maintaining the integrity and focus of our original manuscript.

We hope that these revisions meet your expectations and strengthen the manuscript.

Sincerely,

Ewa Majcherek